# Modulatory Mechanisms of the NLRP3 Inflammasomes in Diabetes

**DOI:** 10.3390/biom9120850

**Published:** 2019-12-09

**Authors:** Sujuan Ding, Sheng Xu, Yong Ma, Gang Liu, Hongmei Jang, Jun Fang

**Affiliations:** 1College of Bioscience and Biotechnology, Hunan Agricultural University, Changsha 410128, Hunan, China; jiayousujuan@hunau.edu.cn (S.D.); mayong@stu.hunau.edu.cn (Y.M.); 2College of Life Sciences, Shandong Agricultural University, Tai’an 271018, Shandong, China; shengseanxu@gmail.com; 3Hunan Provincial Key Laboratory of Animal Nutritional Physiology and Metabolic Process, CAS Key Laboratory of Agro-ecological Processes in Subtropical Region, Institute of Subtropical Agriculture, Chinese Academy of Sciences, National Engineering Laboratory for Pollution Control and Waste Utilization in Livestock and Poultry Production, Changsha 410125, Hunan, China

**Keywords:** diabetes, NLRP3, Glucose tolerance, insulin resistance, gut microbes

## Abstract

The inflammasome is a multiprotein complex that acts to enhance inflammatory responses by promoting the production and secretion of key cytokines. The best-known inflammasome is the NLRP3 (nucleotide-binding oligomerization domain-like receptor [NLR] family pyrin domain-containing 3) inflammasome. The evidence has shown that the NLRP3 inflammasome, IL-1β, thioredoxin-interacting protein (TXNIP), and pyroptosis play vital roles in the development of diabetes. This review summarizes the regulation of type 1 diabetes mellitus (T1DM) and type 2 diabetes mellitus (T2DM) by NLRP3 via modulation of glucose tolerance, insulin resistance, inflammation, and apoptosis mediated by endoplasmic reticulum stress in adipose tissue. Moreover, NLRP3 participates in intestinal homeostasis and inflammatory conditions, and NLRP3-deficient mice experience intestinal lesions. The diversity of an individual’s gut microbiome and the resultant microbial metabolites determines the extent of their involvement in the physiological and pathological mechanisms within the gut. As such, further study of the interaction between the NLRP3 inflammasome and the complex intestinal environment in disease development is warranted to discover novel therapies for the treatment of diabetes.

## 1. Introduction

It has been estimated that there were ~451 million diabetic patients between 18 and 99 years of age in 2017. This figure is expected to exceed 693 million by 2045. Moreover, 20 million people worldwide between the ages of 20 and 99 die of diabetes, and the medical expenses for diabetic patients worldwide have been estimated at $850 billion [1]. Diabetes, characterized by an elevated blood glucose level due to insufficient insulin production, is a heterogeneous disease with multiple causes [2]. Diabetes is a central healthcare concern due to its risk of several severe complications, such as heart disease, stroke, and renal failure. Type 2 diabetes mellitus (T2DM) is the world’s most common metabolic disease and is characterized by insulin secretion defects, which are influenced by lifestyle factors such as age, pregnancy, and obesity and have a strong genetic component [2,3]. Glucose, insulin, lipids, and intestinal microorganisms play important roles in both the diagnosis and treatment of T2DM. Moreover, T2DM is a latent disease that affects people all over the world, with devastating complications such as cardiovascular disease, renal failure, and cancer [4]. In short, T2DM has a hugely negative impact on life, health, and the economy. In contrast to T2DM, T1DM arises due to autoimmune-mediated B cell destruction. It accounts for 5% to 10% of all cases of diabetes mellitus and is usually diagnosed in childhood by the presence of islet cell antibodies, which are absent in T2DM [5]. Patients with T1DM rely on insulin therapy for life [6]. In view of the huge negative effects of diabetes on people and society, it is significant to explore or update the potential treatments for the disease. In this review, we summarized the composition and activation of NLRP3 inflammasomes and their potential therapeutic role in the progression of diabetes.

## 2. The NLRP3 Inflammasome

Inflammasomes are polymorphic complexes formed by pattern recognition receptors activated by various physiological or pathogenic stimuli, which makes them an important component of the innate immune response with the ability to clear pathogens and damaged cells [7]. The NLRP3 (nucleotide-binding and oligomerization domain-like receptor family pyrin domain-containing, NLRP) inflammasome is an innate immune cell sensor that belongs to the NLR family [8]. The NLRP3 inflammasome is considered to be the most characteristic and contains the sensor molecule NLRP3, an apoptosis-associated speck-like protein containing a caspase recruitment domain (CARD) (ASC), and pro-caspase-1 [9]. Two steps are required to complete the activation of NLRP3 inflammasomes in macrophages (Figure 1). First, NLRP3 and pro-IL-1β expression are induced by inflammatory stimuli via NF-κB. NLRP3 inflammasomes are then activated by pathogen-associated molecular patterns (PAMPs) and damage-associated molecular patterns (DAMPs), which leads to the assembly of NLRP3 inflammasomes, the secretion of IL-1β and IL-18, and caspase-1-mediated pyroptosis [10,11]. However, there is strong evidence that lipopolysaccharide (LPS) alone can induce the maturation and production of caspase-1–dependent IL-1β in human monocytes [12]. In addition, researchers could induce activation of the NLRP3 inflammasome directly via the TLR4 signaling pathway without the need for other secondary activators [13]. NLRP3 inflammasome responds to a variety of infectious and endogenous ligands and is involved in a variety of autoimmune diseases, such as obesity [14], diabetes mellitus [15,16], arthritis [17,18], and Alzheimer’s disease [19].

Comprehending the process of NLRP3 activation will aid the development of specific inhibitors, which could enhance the treatment of NLRP3-related diseases. Although the specific regulatory mechanisms of NLRP3 inflammasome activation remains unclear, researchers have found that many stimuli, including some DAMPs and PAMPs, can activate the NLPR3 inflammasome [20,21]. The pathways involved in the activation of the NLPR3 inflammasome are diverse. Changes in ion concentrations, such as reductions in intracellular K^+^, can trigger the activation of NLPR3 [22], whereas the inhibition of Ca^2+^ influx can reduce the activation of the NLRP3 inflammasome [23]. Na^+^ may not be required for activation of the NLRP3 inflammasome, but a study found that Na^+^ influx relies on K^+^ influx in the activation of the NLRP3 inflammasome, and Cl^−^ channel inhibitors (such as flufenamic acid, IAA94, DIDS, and NPPB) may inhibit NLRP3 [22,24,25,26]. In addition, NLRP3 can be activated by the production of reactive oxygen species, lysosomal instability, post-translational modifications of NLRP3, and activation of human caspase-4/5 and mouse caspase-11 [27,28,29,30,31]. The function of NCLX (Na^+^/Ca^2+^ exchanger) expression induced by glucose in rat aortic mesothelial cells indicated that NCLX increased the Ca^2+^ flux of glucose-dependent mitochondria, thus regulating ROS production and subsequent activation of the NLRP3 inflammasome in high glucose conditions. In the initial stage of glucose stimulation, a compensatory increase in NCLX expression was seen to protect mitochondria and preserve endothelial cell function [32]. In addition, studies have shown that obese individuals have elevated caspase activity in monocytes, with palmitate acting to activate caspase-4/5, resulting in the release of inflammatory cytokines, which suggests that caspases may be a novel therapeutic target for the reduction of obesity-related inflammation [33].

## 3. Metabolic Disease and Diabetes

Obesity rates have risen sharply worldwide, reaching epidemic levels, and leading to a significant increase in the prevalence of metabolic diseases [34]. At the same time, a variety of health complications are associated with obesity, including cancer, hypertension, diabetes, cardiovascular disease, and nonalcoholic fatty liver (NAFLD) [35,36]. As such, obesity and obesity-related health complications have received extensive attention and have been the subject of many investigations, including research on prevention, treatment strategies, and potential mechanisms. Diabetes is a metabolic disease characterized by marked hyperglycemia due to defects in insulin secretion and/or insulin action, as well as polyphagia and blurred vision [37]. According to the National Diabetes Data Group (NDDG), the glycemic criteria of diabetes are a fasting blood glucose level of at least 140 mg/dL (7.8 mmol/L) for two consecutive tests, or at least 200 mg/dL (11.1 mmol/L) 2 h after the standard oral glucose tolerance test (75 g anhydrous glucose) [38].

Insulin resistance is a key pathophysiological process that occurs in the liver, muscle, and adipose tissue during the development of type 2 diabetes mellitus (T2DM) and is listed as one of the four major noncommunicable diseases by the WHO [39]. In addition, abnormal lipid metabolism and intestinal microbial dysbiosis are vital factors in the development of diabetes mellitus. Lipid metabolism is a complex physiological process linked to nutrient regulation [40], hormone regulation [41], and homeostasis [42]. The ability to regulate lipid metabolism is critical to maintaining health in both single-celled organisms and humans. However, unhealthy lifestyles and chronic overnutrition in modern lifestyles have led to the occurrence of serious lipid metabolism disorders. Therefore, an increased understanding of the molecular mechanisms that underlie lipid metabolism is urgently needed to combat these diseases.

Furthermore, the composition of the intestinal microbiome has been recognized as an important factor in the progression of metabolic diseases and obesity. Studies have found that patients with obesity and T2DM have a characteristic gut microbiome that may be associated with the transfer of microbes from the gut to tissues, resulting in inflammation [43]. For example, studies have found that fluctuation of the Firmicutes/Bacteroidetes ratio has an impact on obesity and insulin resistance [44]. Meanwhile, bacterial metabolites, such as peptidoglycan and lipopolysaccharide (LPS) can trigger the innate immune system [45]. Under unbalanced conditions, such as in metabolic diseases, the permeability and integrity of the intestinal tract become impaired, resulting in a transfer of bacteria and bacterial metabolites to the surrounding tissue, eventually leading to long-term inflammatory processes. Such long-term chronic inflammation may be detrimental to the action and secretion of insulin [46]. Insulin resistance occurs before the progression of T2DM, with several metabolic markers (such as fasting glucose, glycosylated hemoglobin, and insulin) correlating with the presence of Lactobacillus and Clostridium [47]. Chronic, low-grade inflammation is common in obese patients and those with T2DM [48]. Diet may induce increased intestinal permeability, which may explain the observed increased translocation of endotoxins [49]. It has been found that translocation of toxins, such as endotoxin, derived from Gram-negative bacteria, can cause low-grade inflammation [50]. The plasma LPS levels were elevated in patients with T2DM compared to nondiabetic patients [51]. The elevated levels of LPS entering the liver may affect both the inflammatory signaling pathways and insulin signaling within the liver [52]. Furthermore, insulin is known to promote the phosphorylation of cytoskeletal proteins by activating myosin light streptokinase, which may contribute to increased intestinal permeability [53].

## 4. NLRP3 and Diabetes

### 4.1. Development of Type 1 Diabetes Mellitus (T1DM) Is Regulated by NLRP3

T1DM is a disease caused by defects in insulin synthesis, predominantly due to autoimmune-mediated destruction of pancreatic β-cells. T1DM accounts for 5% to 10% of all cases of diabetes [54]. Generally speaking, T1DM is considered a cellular immune disease [55]. The NLRP3 inflammasome plays a critical role in the progression of insulin resistance during the course of T2DM, but its role in the autoimmune T1DM remains to be investigated [56,57] (Table 1). However, there is increasing evidence that innate immune responses involving Toll-like receptors (TLR) play an important role in the development of T1DM. TLRs are pattern recognition molecules that recognize a pathogen on the surface of immune cells, thus inducing the production of IL-1β [58]. It is believed that IL-1β may be a biomarker for the early development of T1DM [59,60]. Therefore, an increased understanding of the role of IL-1β in the pathophysiology of T1DM may lead to improved treatments for T1DM. Pathogen-associated and injury-related molecular pattern molecules and environmental stimuli are both known to activate NLRP3 [61]. NLRP3 can be activated by pathogen-related and injury-associated molecular pattern molecules. The evidence has revealed that TLRs and IL-1β are both upregulated in monocytes of T1DM [62], and NLRP3 is likely to be activated in proinflammatory status. However, the specific role of NLRP3 in T1DM remains to be explored. Moreover, inflammation-independent pathways, such as neutrophil- and macrophage-derived serine proteinases, activate IL-1β and may play a potential role in T1DM [63,64].

Unlike tissue-resident cells, such as macrophages and dendritic cells, T cells are rarely seen in healthy pancreatic islets. Therefore, the recruitment of effector T cells to islets is a key initiating step in the inflammation and subsequent β cell destruction characteristic of T1DM [16]. Studies have shown that knockout of NLRP3 not only inhibits T cell activation and Th1 cell differentiation, it also inhibits the migration of diabetogenic cells to pancreatic islets by down-regulating the expression of chemotactic genes in islet T cells and non-hematopoietic cells [16]. Moreover, mitochondrial dysfunction and apoptosis can be induced by activating factors, such as ATP, which leads to the release of oxidized mitochondrial DNA (mDNA) into the cell, resulting in activation of NLRP3 [65]. In vivo studies demonstrated that mDNA increased the levels of Th17/Tc17/Th1/Tc1 cells in pancreatic lymph nodes, promoting the development of T1DM, whereas this T1DM development was inhibited in *NLRP3*^−/−^ mice. In addition, mDNA-mediated activation of the NLRP3 inflammasome triggers caspase-1–dependent production of IL-1β and contributes to pathogenic cell responses in streptozotocin-induced T1DM models [66].

### 4.2. The Development of Type 2 Diabetes Mellitus (T2DM) Is Regulated by NLRP3

The prevalence of T2DM is rising globally, and it has become a global health burden due to its numerous complications, including cardiovascular disease and cancer [4]. Therefore, improving our understanding of the pathogenesis of T2DM is crucial to facilitate the development of new therapies. Accumulating evidence highlights the central role of glucose homeostasis, insulin, and lipid metabolism in the pathophysiology of T2DM. The NLRP3 inflammasome plays a significant role in regulating the innate immune system by interacting with TXNIP (thioredoxin-interacting protein) [70]. Moreover, activation of the NLRP3 inflammasome affects glucose tolerance, insulin sensitivity, and interactions with gut microbes [3,71,72]. This section will review the role of NLRP3 in the development of T2DM.

**Glucose tolerance:** NLRP3 promotes IL-1β and IL-18 production. Active caspase-1 hetero-tetramers assembled from p10 and p20 subunits can transform inactive pro-IL-1β and pro-IL-18 into their biologically active secreted forms [73]. However, the physiological role of IL-1β in glucose metabolism is still unknown. The evidence has demonstrated that chronically upregulation of IL-1 β leads to an increase in insulin levels that may be detrimental to metabolism, possibly because insulin enhances the inflammatory status of macrophages by promoting glucose uptake and metabolism and increases the expression of insulin receptors in macrophages of DIO mice. IL-1β was found to improve the absorption of glucose into macrophages, with insulin intensifying the proinflammatory effects via regulation of the insulin receptor, glucose metabolism, and production of reactive oxygen species, and NLRP3 inflammasomes mediate the secretion of IL-1β. Increasing the glucose excretion into the urine can prevent glucose overload in tissues, thus preventing the harmful and negative effects of glucose-induced IL-1β [71].

IL-18 is a proinflammatory cytokine produced by a variety of immune cells such as dendritic cells, macrophages, T cells, and B cells, and it is a member of the IL-1 family of cytokines, initially considered as interferon gamma (IFN-γ)-inducing factor [74,75]. Evidence revealed that IL-18 is related to obesity [76], insulin resistance [77], and dyslipidemia [76]. A study was conducted to investigate whether IL-18 promoter-607 C/a polymorphism affects serum IL-18 concentration and glucose metabolism in Chinese subjects through 232 patients with impaired glucose regulation (IGR) or type 2 diabetes mellitus. The results showed that IL-18 level in IGR or type 2 diabetes mellitus was remarkably increased in comparison that in normal glucose regulation. In addition, Genotype A/A of IL-18 gene promoter -607 C/A polymorphism was related to the prevalence of type 2 diabetes mellitus and the level of blood glucose after 2 h [78].

TXNIP-deficient mice show sensitivity to hyperlipidemia, and the TXNIP gene is located on the 1q21-1q23 chromosome within a T2DM locus [79]. Because TXNIP is the strongest glucose-responsive gene in pancreatic β-cells, TXNIP may be a potential therapeutic target in the treatment of diabetes [79]. The biological relevance of TXNIP dysfunction may be particularly relevant in cases of recurrent hyperglycemia. In addition, TXNIP regulates triglycerides by influencing blood glucose levels. In addition to affecting the expression of TXNIP, hyperglycemia can also induce IL-1β release from islet cells, and these findings prompted a study of the secretion of the TXNIP-NLRP3 inflammasome in islet cells [80]. Inflammatory activators, such as uric acid crystals, induce dissociation of TXNIP from thioredoxin in a ROS-sensitive manner, thereby facilitating binding of TXNIP to NLRP3. Studies have shown that both *TXNIP*
^−/−^ and *NLRP3*
^−/−^ mice exhibit improved glucose tolerance and insulin sensitivity [70]. These findings suggest that TXNIP is involved in the activation of the NLRP3 inflammasome and may provide insight into the involvement of IL-β in the pathogenesis of T2DM.

**Insulin resistance:** For 90% to 95% of patients, T2DM is caused by a progressive deficiency in insulin secretion, leading to a lack of relative insulin secretion in insulin-resistant patients [54]. Excess nutrition promotes insulin resistance, and being overweight/obese is a major risk factor for T2DM [81]. Obesity promotes the initiation of NLRP3 inflammasome formation in diabetic patients [3]. In addition to the many drugs currently used to treat diabetes (e.g., nateglinide, metformin, dipeptidyl peptidase-4 (DPP-4) inhibitors, linagliptin, and α-glycosidase inhibitors) [82,83,84,85,86], there are ongoing efforts to develop new natural drugs to the treatment of diabetes without side effects (Table 2). At the same time, understanding the relevant regulatory mechanisms in the development of the disease is crucial to improving treatments. Accumulating evidence highlights a central role for the NLRP3 inflammasome in obesity-induced insulin resistance [56,87,88]. High levels of IL-1β may contribute to insulin insensitivity in obese individuals. In the adipose tissue of such individuals, especially in macrophages, the expression of the NLRP3 inflammasome components, the activity of caspase-1, and the level of IL-1β are increased, all of which are directly correlated with insulin resistance, metabolic syndrome, and the severity of T2DM [56,88]. The NLRP3 inflammasome activates and promotes the production of IL-1β from pancreatic β-cells and islet-infiltrating macrophages. Moreover, the NLRP3 inflammasome can be activated by metabolic signaling molecules such as glucose, saturated fatty acids (SFA), and uric acid during obesity, leading to the production of IL-1β and cytokines [70,89]. Dendritic cells exposed to an SFA-rich high-fat diet (HFD) showed significant infiltration into adipose, and these HFD-derived dendritic cells reduced the insulin sensitivity of adipocytes. This study confirmed that SFA could act as a metabolic trigger to initiate inflammasome formation and promote inflammation and insulin resistance in adipocytes. SFA was found to have a direct effect on the activation of inflammasome through TLR4 in vitro experiments [89]. The influential role of NLRP3 in insulin resistance has been studied in animal models and human adipose tissue samples. Such studies have found that *NLRP3*^−/−^ mice have enhanced glucose tolerance and insulin sensitivity, which may be related to the involvement of TXNIP in inflammasome activation [70].

IL-18 has been found to be involved in the regulation of metabolic homeostasis and insulin resistance; however, IL-18 has been found to be necessary for the prevention of hyperphagia in IL-18–deficient mice models [90,91]. Evidence suggests that STAT3-activated cytokines via transmembrane receptor signaling can activate the AMP-activated kinase (AMPK) signaling pathway, which enhances fat oxidation in skeletal muscle, and sequentially reduce insulin resistance induced by a high-fat diet (HFD) [92]. Obesity is known to cause low-grade chronic inflammation [93], and elevated IL-18 levels have been found in obese and type 2 diabetics [94,95]. So what is the role of il-18 in this? The study showed that IL-18 receptor^−/−^ mice exhibit weight gain, ectopic lipid deposition, inflammation, and attenuated AMPK signaling in skeletal muscle, suggesting that IL-18 is involved in metabolic homeostasis, inflammation, and insulin resistance by activating the mechanism of adenosine monophosphate activated protein kinase (AMPK) in skeletal muscle [77].

## 5. NLRP3 and Adipose Tissue

Adipose tissue is the site of the integration of physiological status, energy balance, and glucose homeostasis [103]. Individuals with metabolic diseases, such as diabetes and obesity often experience chronic inflammation and adipose tissue dysfunction [104]. Adipose tissue damaged by hyperglycemia is characterized by oxidative stress and macrophage infiltration, which further exacerbates the degree of inflammation within the adipose tissue [105]. Evidence has shown that endoplasmic reticulum (ER) stress correlates with inflammation in adipose tissue [106]. Insulin-producing islet β-cells are able to accommodate high-speed transport of cargo proteins through the ER [107]. Studies have demonstrated that NLRP3-mediated inflammation and cell death can be caused by ER stress [108]. The ER responds to the accumulation of intraluminal unfolded proteins by activating intracellular signal transduction pathways, leading to programmed cell death in what is termed the unfolded protein response (UPR) [109]. In diabetic individuals, high glucose levels lead to excessive production of ROS and hyperpolarization of mitochondria via increased metabolic input to the mitochondria. High levels of ROS can activate the UPR pathway and may result in an inflammatory response [110]. In this vicious circle of inflammation, oxidative stress exacerbates ER stress, leading to the destruction of mitochondrial morphology and function [111]. In addition, ROS can induce mitochondrial division mediated by dynamin-related protein1 (Drp1) [112]. Studies have revealed that metformin and resveratrol can maintain the integrity of mitochondria via suppression of Drp1 activity and inhibition of NLRP3 inflammasome activation by preventing ER stress, thereby protecting adipose from damage mediated by a high glucose level [113].

## 6. NLRP3 and the Microbiome

Symbiotic bacteria play a crucial role in maintaining normal immune function. Metabolic disorder–related diseases such as obesity, diabetes, and cardiovascular diseases are closely related to alteration in the composition and function of the gut microbiome [47,114]. An analysis of the composition and function of the intestinal microbiome in 15 children with T1DM, 15 children with maturity-onset diabetes of the young 2 (MODY2), and 13 healthy children showed that the diversity and composition of the intestinal microbiome in T1DM and MODY2 was decreased relative to healthy controls. For example, there was an increased abundance of *Ruminococcus* and *Bacteroides* and a decreased abundance of *Bifidobacterium* and *Faecalibacterium* in T1DM; the relative abundance of *Prevotella* was increased in MODY2, but *Ruminococcus* and *Bacteroides* were reduced. Moreover, intestinal permeability was increased in MODY2 and T1DM, accompanied by increased serum proinflammatory cytokines (e.g., IL-1β, IL-6, and TNF-α) and LPS in T1DM [115]. The inflammasome complexes NLRP3 is a multiprotein complex that recognizes microbial-associated molecular patterns and participates in proinflammatory pathways, and the mice lack these complexes show altered intestinal microbial composition and lead to NAFLD [116]. Moreover, the study found that the expression of IL-1β and NLRP3 mRNA was increased in monocyte-derived macrophages (MDMs) derived from patients with a new diagnosis of T2DM after LPS stimulation in comparison with healthy MDMs [117]. It has been reported that NLRP3 promotes the secretion of antimicrobial peptides in the intestinal epithelium by promoting the production of more IL-1β than IL-18, leading to changes in the microbiome composition [118]. IL-18 is secreted by epithelial cells to stimulate the barrier function and regeneration of epithelial cells, and the activation of inflammasome has a proinflammatory effect [119]. NLRP3-deficient mice had altered interactions between the intestinal microbiome and the host, which may influence the progression of symptoms associated with metabolic syndromes. Furthermore, low-grade intestinal lesions were present in these NLRP3-deficient mice that depended on excessive growth of Prevotellaceae and Bacteroidetes [116], and the ratio of Firmicutes to Bacteroidetes was decreased [120]. CCL5 is caused by bacterial and viral infections and recruits a variety of innate and adaptive immune cells by activating toll-like receptors on epithelial cells [121]. The gut microbiota in mice with NLRP3 inflammasome-deficient mice induced colitis by epithelial CCL5 secretion [119]. Unfortunately, the extent to which the NLRP3 inflammasome is involved in the diabetic intestinal tract and the specific mechanisms by which it participates and maintains the intestinal homeostasis via interactions with the intestinal microbiome remains to be explored.

## 7. Conclusions and Future Perspective

In view of the prevalence of diabetes mellitus, both T1DM and T2DM, new treatment options are urgently needed. The NLRP3 inflammasome provides a platform for the production of IL-1β and IL-18. Following the onset of NLRP3-mediated inflammation, cells secrete a large number of proinflammatory cytokines, which aggravates insulin resistance and accelerates the progression of the disease. NLRP3 inflammasome-induced IL-1β production plays an important role in the development of obesity and diabetes. IL-1β directly inhibits the insulin signaling pathway by reducing tyrosine phosphorylation of insulin receptor substrate-1 (IRS-1) and negatively regulating insulin receptor substrate-1 (IRS-1) gene expression. In addition, the NLRP3 inflammasome participates in the inflammation and glucose homeostasis by participating in immune regulation of adipose tissue. Meanwhile, intestinal microbes actively participate in the development of diabetes, with the intestinal microbiota possessing the ability to affect the response of cells to insulin. Butyric acid produced by intestinal microbes could improve human insulin sensitivity, whereas propionic acid increased the risk of T2DM [72]. Furthermore, some studies have found that microbe-derived imidazole propionate hinders insulin signal transduction via mechanistic target of rapamycin complex 1 (mTORC1) [122]. During the pathogenesis of diabetes mellitus, the interactions between the NLRP3 inflammasome and intestinal microbes/microbial metabolites, and how these interactions influence and maintain intestinal homeostasis, remain to be explored. Moreover, many studies are carried out to find potential new therapies for diabetes. An important challenge we now need to face is how to translate the findings of in vitro and animal experiments into humans. After all, there is a large gap between in vitro and in vivo experiments that needs bridging, in addition to the differences in drug responses between animals and humans.

## Figures and Tables

**Figure 1 biomolecules-09-00850-f001:**
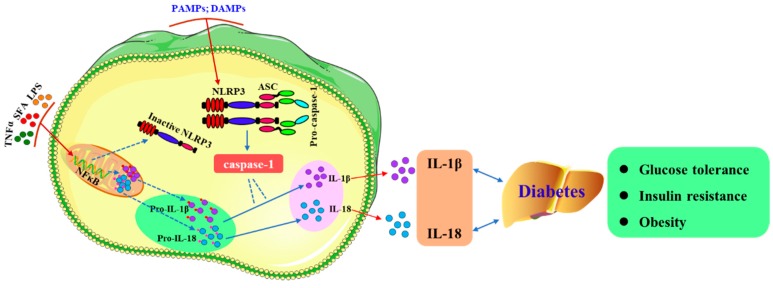
Activation of NLRP3 and secretion of IL-1β and IL-18. NLRP3 inflammasomes are composed of the sensor molecule NLRP3, an apoptosis-associated speck-like protein containing a caspase recruitment domain (CARD) (ASC), and pro-caspase-1. There are two steps to activate the NLRP3 inflammasome in macrophages. First, NF-κB expression is induced by inflammatory stimuli (such as TNFα, SFA, or LPS) resulting in the expression of pro-IL-1β and pro-IL-18; Second, caspase-1 mediates the maturation and secretion of IL-1β and IL-18 following activation of the inflammasome by PAMPs and DAMPs. After NLRP3 is activated, cells secrete a large number of proinflammatory cytokines (e.g., IL-1β and IL-18), which aggravate glucose tolerance, insulin tolerance, and the progression of diabetes. SFA: Saturated fatty acids.

**Table 1 biomolecules-09-00850-t001:** NLRP3 participated in the regulation of diabetes in different models.

Type of Diabetes	Model	NLRP	Functional Consequences	Reference
T1DM	PBMCs; GCs	NLRP3; NLRP1	Progression of the disease	[67]
T1DM	*NLRP3*^−/−^ mice	NLRP3	Suppressed T-cell activation and modulated pathogenic T-cell migration to the pancreatic islets via regulating the expression of chemokine receptors CCR5 and CXCR3 by NLRP3	[16]
T1DM	*NLRP3*^−/−^ mice	NLRP3	Increased the myeloid-derived suppressor cell and mast cell numbers of pancreatic lymph nodes in conjunction with an ascent of the IL-6, IL-10, and IL-4 in pancreatic tissue of NLRP3-deficient mice	[66]
T1DM	THP-1 cells	NLRP3	Promoted sequestration into phagophores	[68]
T1DM/T2DM	Wild-type mice	NLRP3/ASC	Progression of the disease	[69]
T1DM	*NLRP3*^−/−^ mice	NLRP3	Reduced glucose tolerance and insulin sensitivity	[70]

PBMCs: blood mononuclear cells; GCs: granulocytes.

**Table 2 biomolecules-09-00850-t002:** Regulation of insulin by NLRP3 in the progression of diabetes mellitus.

Subjects	Model	Regulatory Mechanism	Functional Consequences	Reference
Betaine	Hep 2 cells, db/db mice	Decreased the production of IL-1β via interactions with FOXO1 and TXNIP and inhibited the activation of the NLRP3 inflammasome	Inhibition of RS-induced activation of the NLRP3 inflammasome in diabetic livers	[96]
Mangiferin	Perivascular adipose tissue	Increased LKB1-dependent AMPK activity; inhibited NLRP3 inflammasome activation; reduced the secretion of IL-1β	Prevention of endothelial insulin resistance	[97]
Berberine	ob/ob mice	Activation of AMPK-dependent autophagy in adipose tissue-resident macrophages	Alleviation of insulin resistance	[98]
MUFA	Mice; Bone marrow-derived cells	Decreased the formation of pro-IL-1β, reduced the secretion of IL-1β and maintained the activation of adipose AMPK	Improved insulin resistance mediated by IL-1β and adipose dysfunction	[99]
SFA	Mice; BMDC;	Improved the expression of IL-1R1, TLR4, and caspase-1 and increased the secretion of IL-1β	Decreased insulin levels in adipose	[89]
WMW	HepG2 cells	Regulated the downstream insulin signaling pathway via reduced IR and IRS-1; decreased IL-1β and NFκB	Alleviation of insulin resistance	[100]
CBX	Mice	Suppressed the IκB-α/NF-κB pathway and inhibited the activation of the NLRP3 inflammasome	Alleviation of insulin resistance in the liver and skeletal muscle	[101]
Silymarin	Hnf-1α- knockout mice	Reduced expression of IL-1β mediated by HG and inflammasome	Alleviation of insulin resistance	[102]

RS: reactive species; MUFA: Monounsaturated Fatty Acid; SFA: saturated fatty acids; BMDC: bone marrow-derived dendritic cells; WMW: Wu-Mei-Wan, a Chinese herbal formula; IR: insulin receptor; IRS-1: insulin receptor substrate-1; CBX: Carbenoxolone; HG: high glucose.

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
