# Peer review of "Modulatory Mechanisms of the NLRP3 Inflammasomes in Diabetes"

_biomolecules, 2019, doi:10.3390/biom9120850_

Round 1
Reviewer 1 Report
1st comment: The point was that NLRP3 would not need to be presesnted in the introduction since it is followed by own chapter on NLRP3.
2nd comment: Revision is not accurate. Caspase-1 is needed for both pyroptosis (gasdermin D as a substrate) and IL-1b activation.
Comments on the term "affect" refer to its meaningless content. It should not be used but replaced by a specific characterization (is something increased/reduced etc.). "Affect" does not tell anything.
Reviewer 2 Report
The review describes activation of the NLRP3 inflammasome and the role of NLRP3 in diabetes (in the context of interleukin-1beta production).
The authors have addressed the points raised by the reviewer sufficiently and appropriately.
Author Response
Please see the attachment.

This manuscript is a resubmission of an earlier submission. The following is a list of the peer review reports and author responses from that submission.
Round 1
Reviewer 1 Report
This is a confusing manuscript about NLRP3 inflammasome and diabetes. Please, find some detailed comments below.
The manuscript needs a careful language revision taking into account also meanings. See for example, "Diabetes... is a heterogenous disease with multiple military services" (Page 2, lines 50-51); "The whole suggest.." ? (page 7, line 233) "...that TXNIP involved in NLRP3 inflammasome activation may be an effective way to monitor the involvement of IL-1b in the pathogenesis of T2DM". Please, specify, how. Introduction should be more focused. At the beginning of introduction, it should be specified whether T1D or T2D is concerned. In the abstract, T1D is emphasized and it sounds unrealistic that half of them would have remained undiagnosed. Authors report that T1D "is related to the presence of islet cell antibodies" (Page 2, lines 55-56) for which the inflammasome activation should be linked to that in order to be "decisive" (abstract). Tables are unclear. Please, avoid terms, such as "change", "decrease", or "increase" when there is nothing to be compared with. There are unclear sentences also in text. Please, see e.g. "IL-1b the absorption of glucose by macrophages, and insulin enhanded the pro-inflammatory model through insulin receptor, glucose metabolism, production of reactive oxygen species and NLRP3 inflammasome mediated IL-1b secretion... (Page 6, lines 217-219). It remains unclear how TXNIP mutations would be used in therapy of diabetes (page 7, lines 224-225). Figure 1 is not tightly related to the manuscript text.Author Response
1.The manuscript needs a careful language revision taking into account also meanings. See for example, "Diabetes... is a heterogenous disease with multiple military services" (Page 2, lines 50-51); "The whole suggest.." ? (page 7, line 233) "...that TXNIP involved in NLRP3 inflammasome activation may be an effective way to monitor the involvement of IL-1b in the pathogenesis of T2DM".
Response: The issue has been revised, as following:
Line 49-50: Diabetes, characterized by an elevated blood glucose level due to insufficient insulin production, is a heterogeneous disease with multiple causes.
Line 259-261: These findings suggest that TXNIP is involved in the activation of the NLRP3 inflammasome and may provide insight into the involvement of IL-β in the pathogenesis of T2DM.
2. Please, specify, how. Introduction should be more focused. At the beginning of introduction, it should be specified whether T1D or T2D is concerned.
Response: We have revised this issue, as following:
Line 52-63: T2DM is the world’s most common metabolic disease and is characterized by insulin secretion defects, which are influenced by lifestyle factors such as age, pregnancy, and obesity and have a strong genetic component [2,3]. Glucose, insulin, lipids, and intestinal microorganisms play important roles in both the diagnosis and treatment of T2DM. Moreover, T2DM is a latent disease that affects people all over the world, with devastating complications such as cardiovascular disease, renal failure, and cancer [4]. In short, T2DM has a hugely negative impact on life, health, and the economy. In contrast to T2DM, T1DM arises due to autoimmune-mediated B cell destruction. It accounts for 5% to 10% of all cases of diabetes mellitus and is usually diagnosed in childhood by the presence of islet cell antibodies, which are absent in T2DM [5]. Patients with T1DM rely on insulin therapy for life [6].
3. In the abstract, T1D is emphasized and it sounds unrealistic that half of them would have remained undiagnosed.
Response: We have revised it in Line 49.
4. Authors report that T1D "is related to the presence of islet cell antibodies" (Page 2, lines 55-56) for which the inflammasome activation should be linked to that in order to be "decisive" (abstract).
Response: We have revised it, as following:
Line 256-261: Inflammatory activators, such as uric acid crystals, induce dissociation of TXNIP from thioredoxin in a ROS-sensitive manner, thereby facilitating binding of TXNIP to NLRP3. Studies have shown that both TXNIP –/– and NLRP3 –/– mice exhibit improved glucose tolerance and insulin sensitivity [73]. These findings suggest that TXNIP is involved in the activation of the NLRP3 inflammasome and may provide insight into the involvement of IL-β in the pathogenesis of T2DM.
5. Tables are unclear. Please, avoid terms, such as "change", "decrease", or "increase" when there is nothing to be compared with.
Response: We have revised the Table 1 and Table 2, as following:
Line 221: Table 1. Effects of NLRP3 during the development of diabetes.
Line 291: Table 2. Regulation of insulin by NLRP3 in the progression of diabetes mellitus.
6. There are unclear sentences also in text. Please, see e.g. "IL-1β the absorption of glucose by macrophages, and insulin enhanded the pro-inflammatory model through insulin receptor, glucose metabolism, production of reactive oxygen species and NLRP3 inflammasome mediated IL-1b secretion... (Page 6, lines 217-219).
Response: We have revised this sentences in the text, as following:
Line 241-245: IL-1β was found to improve the absorption of glucose into macrophages, with insulin intensifying the pro-inflammatory effects via regulation of the insulin receptor, glucose metabolism, and production of reactive oxygen species. Meanwhile, the NLRP3 inflammasome was also found to mediate IL-1β secretion.
7. It remains unclear how TXNIP mutations would be used in therapy of diabetes (page 7, lines 224-225).
Response: The TXNIP mutations may be particularly relevant to recurrent episodes of hyperglycemia and regulate triglycerides by influencing blood glucose levels. Besides affecting the expression of TXNIP, hyperglycemia can induce islet cells to release IL-1β, which prompted the study of the treatment and secretion of TXNIP-NLRP3 inflammasome in islet cells. Moreover, the study shown that both TXNIP –/– and NLRP3 –/– exhibit better glucose tolerance and insulin sensitivity. We have revised it, as following:
Line 253-261: In addition to affecting the expression of TXNIP, hyperglycemia can also induce IL-1β release from islet cells. These findings prompted a study of the secretion of the TXNIP-NLRP3 inflammasome in islet cells [78]. Inflammatory activators, such as uric acid crystals, induce dissociation of TXNIP from thioredoxin in a ROS-sensitive manner, thereby facilitating binding of TXNIP to NLRP3. Studies have shown that both TXNIP –/– and NLRP3 –/– mice exhibit improved glucose tolerance and insulin sensitivity [73]. These findings suggest that TXNIP is involved in the activation of the NLRP3 inflammasome and may provide insight into the involvement of IL-β in the pathogenesis of T2DM.
8. Figure 1 is not tightly related to the manuscript text.
Response: We have revised this issue in Line 87-96 and the legend of Figure 1 has been added in Line 123-132:
Line 87-96: Two steps are required to complete the activation of NLRP3 inflammasomes in macrophages (Figure 1). First, NLRP3 and pro-IL-1β expression is induced by inflammatory stimuli via NF-κB. NLRP3 inflammasomes are then activated by pathogen-associated molecular patterns (PAMPs) and damage-associated molecular patterns (DAMPs), which leads to the assembly of NLRP3 inflammasomes, the secretion of IL-1β and IL-18, and caspase-1-mediated pyroptosis [13,14]. However, there is strong evidence that lipopolysaccharide (LPS) alone can induce the maturation and production of caspase-1–dependent IL-1β in human monocytes [15]. In addition, researchers could induce activation of the NLRP3 inflammasome directly via the TLR4 signaling pathway without the need for other secondary activators [16].
Line 123-132: Figure 1. Activation of NLRP3 and secretion of IL-1β and IL-18. NLRP3 inflammasome consists of the sensor molecule NLRP3, apoptosis-associated speck-like protein containing a CARD (ASC), and pro-caspase-1. There are two steps to activate the NLRP3 inflammasome in macrophages. First, NF-κB expression is induced by inflammatory stimuli (such as TNFα, SFA, or LPS) resulting in the expression of pro-IL-1β and pro-IL-18; Second, caspase-1 mediates the maturation and secretion of IL-1β and IL-18 following activation of the inflammasome by PAMPs and DAMPs. After NLRP3 is activated, cells secrete a large number of pro-inflammatory cytokines (e.g. IL-1β and IL-18), which aggravate glucose tolerance, insulin tolerance and the progression of diabetes. SFA: Saturated fatty acids.
Reviewer 2 Report
In this manuscript, Ding et al reviewed the important roles of NLRP3 inflammasome in type 1 and type 2 diabetes. The authors tried to summarize the data regarding the prevalence of type 1 and type 2 diabetes and the involvement of NLRP3 inflammasome in these diseases. At the end, the authors pointed an interesting direction for the future research--- investigate the interactions between gut microbiota and activation of the NLRP3 inflammasome in the setting of diabetes. Overall, the review presented in the manuscript is helpful for anyone who is interested in studying the roles of NLRP3 inflammasome in diabetes. However, the manuscript could be improved with following changes.
Language: Language editing is highly recommended. Currently, there are many grammar errors and broken sentences in the manuscript. For example, line 72, “cells was promoted…”, line 217 “IL-1b the absorption of glucose by macrophages, and insulin enhanced the proinflammatory model…”, Because of these language issues, sometimes it is difficult to understand what the authors tried to deliver. Abbreviations: The authors do not always provide the full name of abbreviations when abbreviations are introduced for the first time. For example, line 33 “ER-stress”, line 88, “PAMPs and DAMPs”. Line 334, “IRS-1”. Another problem is the authors do not provide the same full name for abbreviations. For example, they provided full name of ASC in line 63, line 69, and line 84 but the full names are not consistent. Structure: In the “1 introduction” 2nd paragraph, the authors briefly discussed about NLRP3 inflammasome. However, it is highly overlapped with the discussion in the “2 NLRP3 inflammasomes”. In the “7 conclusion and future perspective”, the authors discussed how IL1b signaling inhibited insulin pathway. Should such information be discussed earlier when the authors discuss about the relationship of inflammasome and insulin resistance? Figure 1 needs detailed figure legend. Regarding NLRP3 inflammasome, I have not seen any other papers call them as “NLPR3 inflammatory bodies” (line 87, 343), “NLRP3 inflammatory corpuscle” (line 100), “NLRP3 inflammatory” (line 88). Please be consistent. Sometimes, I am not sure if the authors were referring inflammasome or inflammatory. For example, line 181, 183, “non-inflammatory independent pathways…” Please describe other people’s research more precisely and accurately. line 256 “dendritic cells derived from high-fat diets…” I don’t think dendritic cells could derive from high fat diet. Line 78, “various physiological or pathogenic pathogens”. How could a pathogen be “physiological pathogen”? In the line 151-163, yes, obesity and T2DM are associated with changed composition of gut microbiota and increased gut permeability. However, the way the authors describe was not very accurate. Please rephrase. In the line 223-235, please discuss the relation between TXNIP and NLRP3 inflammasome more clearly. Line 223, “in TXNIP-deficient mice, TXNIP is located on ….” This sentence doesn’t make sense to me. Line 258, please add a reference when the authors discuss dendritic cells. For line 277-280, two sentences basically repeat the same information. Line 317, “NLRP3-deficient inflammasome…” also doesn’t make sense. Line 300 “structure and function of the gut microbiota”. I am not sure what the authors mean “structure of the gut microbiota”Author Response
Reviewer 2:
In this manuscript, Ding et al reviewed the important roles of NLRP3 inflammasome in type 1 and type 2 diabetes. The authors tried to summarize the data regarding the prevalence of type 1 and type 2 diabetes and the involvement of NLRP3 inflammasome in these diseases. At the end, the authors pointed an interesting direction for the future research--- investigate the interactions between gut microbiota and activation of the NLRP3 inflammasome in the setting of diabetes. Overall, the review presented in the manuscript is helpful for anyone who is interested in studying the roles of NLRP3 inflammasome in diabetes. However, the manuscript could be improved with following changes.
1. Language: Language editing is highly recommended. Currently, there are many grammar errors and broken sentences in the manuscript. For example, line 72, “cells was promoted…”, line 217 “IL-1b the absorption of glucose by macrophages, and insulin enhanced the proinflammatory model…”, Because of these language issues, sometimes it is difficult to understand what the authors tried to deliver.
Response: The langue in the manuscript has been edited by AH Editing company (https://ahediting.com/).
2. Abbreviations: The authors do not always provide the full name of abbreviations when abbreviations are introduced for the first time. For example, line 33 “ER-stress”, line 88, “PAMPs and DAMPs”. Line 334, “IRS-1”. Another problem is the authors do not provide the same full name for abbreviations. For example, they provided full name of ASC in line 63, line 69, and line 84 but the full names are not consistent.
Response: We have revised the full name of abbreviations for the first time in the text, as following:
Line 32: endoplasmic reticulum stress;
Line 90-91: pathogen-associated molecular patterns (PAMPs) and damage-associated molecular patterns (DAMPs);
Line 356: insulin receptor substrate-1 (IRS-1);
Line 70-71: apoptosis-associated speck-like protein containing a CARD (ASC).
3. Structure: In the “1 introduction” 2nd paragraph, the authors briefly discussed about NLRP3 inflammasome. However, it is highly overlapped with the discussion in the “2 NLRP3 inflammasomes”.
Response: We have revised this section, as following:
Line 64-77: NLRP3 is a member of the nucleotide-binding oligomeric domain-like receptor (NLR) family. An important feature of NLR family members is that they carry a central nucleotide binding domain, and most possess a C-terminal leucine-rich repetitive domain (LRR) and a variable N-terminal domain [7]. The NLR family members are intracellular immune cell sensors involved in cytokine secretion and innate immune responses [8]. The NLRP3 inflammasome is considered to be the most characteristic and contains adapter molecules, apoptosis-associated speck-like protein containing a CARD (ASC), and pro-caspase-1 [9]. NLRP3 was initially linked to hereditary autoinflammatory syndrome [7,10]. Activation of NLRP3 promotes the secretion of a large number of proinflammatory cytokines from cells, whereas caspase-1 induces a form of cell death termed pyroptosis, which elicits an IL-1–mediated immune response [9]. IL-1β is an efficient proinflammatory cytokine that is elevated within pancreatic circulation and islets during the development of T2DM in obese patients, which implicates it as an important driver of this disease [11].
4. In the “7 conclusion and future perspective”, the authors discussed how IL1b signaling inhibited insulin pathway. Should such information be discussed earlier when the authors discuss about the relationship of inflammasome and insulin resistance?
Response: The relationship of inflammasome and insulin resistance has been discussed in the Line 273-281, as following:
“High levels of IL-1β may contribute to insulin insensitivity in obese individuals. In the adipose tissue of such individuals, especially in macrophages, the expression of the NLRP3 inflammasome components, the activity of caspase-1, and the level of IL-1β are increased, all of which are directly correlated with insulin resistance, metabolic syndrome, and the severity of T2DM [59,86]. The NLRP3 inflammasome activates and promotes the production of IL-1β from pancreatic β-cells and islet-infiltrating macrophages. Moreover, the NLRP3 inflammasome can be activated by metabolic signaling molecules such as glucose, SFA, and uric acid during obesity, leading to the production of IL-1β and cytokines”.
5. Figure 1 needs detailed figure legend. Regarding NLRP3 inflammasome, I have not seen any other papers call them as “NLPR3 inflammatory bodies” (line 87, 343), “NLRP3 inflammatory corpuscle” (line 100), “NLRP3 inflammatory” (line 88). Please be consistent. Sometimes, I am not sure if the authors were referring inflammasome or inflammatory. For example, line 181, 183, “non-inflammatory independent pathways…”
Response: The NLRP3 inflammasome has been consistent in the whole manuscript. And “non-inflammatory independent pathways…” is correct, which means an pathways. The legend of Figure 1 has been added in the text, as following:
Line 123-132: Figure 1. Activation of NLRP3 and secretion of IL-1β and IL-18. NLRP3 inflammasome consists of the sensor molecule NLRP3, apoptosis-associated speck-like protein containing a CARD (ASC), and pro-caspase-1. There are two steps to activate the NLRP3 inflammasome in macrophages. First, NF-κB expression is induced by inflammatory stimuli (such as TNFα, SFA, or LPS) resulting in the expression of pro-IL-1β and pro-IL-18; Second, caspase-1 mediates the maturation and secretion of IL-1β and IL-18 following activation of the inflammasome by PAMPs and DAMPs. After NLRP3 is activated, cells secrete a large number of pro-inflammatory cytokines (e.g. IL-1β and IL-18), which aggravate glucose tolerance, insulin tolerance and the progression of diabetes. SFA: Saturated fatty acids.
6. Please describe other people’s research more precisely and accurately. line 256 “dendritic cells derived from high-fat diets…” I don’t think dendritic cells could derive from high fat diet.
Response: This sentence has been re-wrote and shown as following:
Line 281-286: Dendritic cells exposed to an SFA-rich high-fat diet (HFD) showed significant infiltration into adipose, and these HFD-derived dendritic cells reduced the insulin sensitivity of adipocytes. This study confirmed that SFA can act as a metabolic trigger to initiate inflammasome formation and promote inflammation and insulin resistance in adipocytes. SFA was found to have a direct effect on the activation of inflammasome through TLR4 in vitro experiments
7. Line 78, “various physiological or pathogenic pathogens”. How could a pathogen be “physiological pathogen”?
Response: We have changed “various physiological or pathogenic pathogens” to “various physiological or pathogenic stimuli” in Line 81.
8. In the line 151-163, yes, obesity and T2DM are associated with changed composition of gut microbiota and increased gut permeability. However, the way the authors describe was not very accurate. Please rephrase.
Response: We have rephrased this section, as following:
Line 170-183: Insulin resistance occurs before the progression of T2DM, with several metabolic markers (such as fasting glucose, glycosylated hemoglobin, and insulin) correlating with the presence of Lactobacillus and Clostridium [50]. Chronic, low-grade inflammation is common in obese patients and those with T2DM [51]. Diet may induce increased intestinal permeability, which may explain the observed increased translocation of endotoxins [52]. It has been found that translocation of toxins, such as endotoxin, derived from Gram-negative bacteria can cause low-grade inflammation [53]. The plasma LPS levels were elevated in the patients with T2DM compared to nondiabetic patients [54]. The elevated levels of LPS entering the liver may affect both the inflammatory signaling pathways and insulin signaling within the liver [55]. Furthermore, insulin is known to promote the phosphorylation of cytoskeletal proteins by activating myosin light streptokinase, which may contribute to increased intestinal permeability [56].
9. In the line 223-235, please discuss the relation between TXNIP and NLRP3 inflammasome more clearly. Line 223, “in TXNIP-deficient mice, TXNIP is located on ….” This sentence doesn’t make sense to me.
Response: We have re-wrote this sentence, as following:
Line 248-249: TXNIP-deficient mice show sensitivity to hyperlipidemia and the TXNIP gene is located on the 1q21-1q23 chromosome, within a T2DM locus.
10. Line 258, please add a reference when the authors discuss dendritic cells.
Response: We have added a reference in this section in Line 286.
11. For line 277-280, two sentences basically repeat the same information.
Response: We have revised this sentence, as following:
Line 302-303: Evidence has shown that endoplasmic reticulum (ER) stress correlates with inflammation in adipose tissue [98].
12. Line 317, “NLRP3-deficient inflammasome…” also doesn’t make sense.
Response: We have re-wrote this sentence, as following:
Line 339-343: NLRP3-deficient mice had altered interactions between the intestinal microbiome and the host, which may influence the progression of symptoms associated with metabolic syndromes. Furthermore, low-grade intestinal lesions were present in these NLRP3-deficient mice that depended on an excessive growth of Prevotellaceae and Bacteroidetes.
13. Line 300 “structure and function of the gut microbiota”. I am not sure what the authors mean “structure of the gut microbiota”
Response: We have changed “structure and function of the gut microbiota” to “composition and function of the gut microbiome” in Line 323-324.

Reviewer 3 Report
The review describes activation of the NLRP3 inflammasome and the role of NLRP3 in diabetes (in the context of interleukin-1beta production).
Whilst the review details that NLRP3 has a role in intestinal homeostasis and inflammatory pathology, the network and interlinking of physiological, pathological and modulatory mechanisms are not evident. Moreover, at several points within the manuscript, the authors state the activation of the NLRP3 inflammasome promotes the production of IL-1beta - however, do not make any reference to interleukin-18 in the context of NLRP3 and diabetes, which is also a known product of NLRP3 inflammasome activation. The authors should incorporate discussion and evaluation on the implications of excessive IL-1beta and IL-18 production in modulating diabetes, regulated via activation of the NLRP3 inflammasome. Whilst figure 1 provides a good representative schematic of how activation of the NLRP3 inflammasome leads to the secretion of IL-1beta and IL-18, perhaps a second figures demonstrating this in the context of NLRP3 and diabetes would increase the cohesiveness. Lastly, can the authors please detail how their review differs from PMID: 23483669?Author Response
Reviewer 3:
The review describes activation of the NLRP3 inflammasome and the role of NLRP3 in diabetes (in the context of interleukin-1beta production).
1. Whilst the review details that NLRP3 has a role in intestinal homeostasis and inflammatory pathology, the network and interlinking of physiological, pathological and modulatory mechanisms are not evident.
Response: We have revised this issue in Line 169-186:
Line 166-183: Under unbalanced conditions, such as in metabolic diseases, the permeability and integrity of the intestinal tract becomes impaired, resulting in a transfer of bacteria and bacterial metabolites to the surrounding tissue, eventually leading to long-term inflammatory processes. Such long-term chronic inflammation may be detrimental to the action and secretion of insulin [49]. Insulin resistance occurs before the progression of T2DM, with several metabolic markers (such as fasting glucose, glycosylated hemoglobin, and insulin) correlating with the presence of Lactobacillus and Clostridium [50]. Chronic, low-grade inflammation is common in obese patients and those with T2DM [51]. Diet may induce increased intestinal permeability, which may explain the observed increased translocation of endotoxins [52]. It has been found that translocation of toxins, such as endotoxin, derived from Gram-negative bacteria can cause low-grade inflammation [53]. The plasma LPS levels were elevated in the patients with T2DM compared to nondiabetic patients [54]. The elevated levels of LPS entering the liver may affect both the inflammatory signaling pathways and insulin signaling within the liver [55]. Furthermore, insulin is known to promote the phosphorylation of cytoskeletal proteins by activating myosin light streptokinase, which may contribute to increased intestinal permeability [56].
2. Moreover, at several points within the manuscript, the authors state the activation of the NLRP3 inflammasome promotes the production of IL-1beta - however, do not make any reference to interleukin-18 in the context of NLRP3 and diabetes, which is also a known product of NLRP3 inflammasome activation. The authors should incorporate discussion and evaluation on the implications of excessive IL-1beta and IL-18 production in modulating diabetes, regulated via activation of the NLRP3 inflammasome.
Response: We have revised this issue, as following:
Line 217-220: In addition, mDNA-mediated activation of the NLRP3 inflammasome triggers caspase-1–dependent production of IL-1β and contributes to pathogenic cell responses in streptozotocin-induced T1DM models [69].
Line 241-247: IL-1β was found to improve the absorption of glucose into macrophages, with insulin intensifying the pro-inflammatory effects via regulation of the insulin receptor, glucose metabolism, and production of reactive oxygen species. Meanwhile, the NLRP3 inflammasome was also found to mediate IL-1β secretion. Increasing the glucose excretion into urine can prevent glucose overload in tissues, thus preventing the harmful and chronic effects of glucose-induced IL-1β [74].
Line 252-256: In addition, TXNIP regulates triglycerides by influencing blood glucose levels. In addition to affecting the expression of TXNIP, hyperglycemia can also induce IL-1β release from islet cells. These findings prompted a study of the secretion of the TXNIP-NLRP3 inflammasome in islet cells.
3. Whilst figure 1 provides a good representative schematic of how activation of the NLRP3 inflammasome leads to the secretion of IL-1beta and IL-18, perhaps a second figures demonstrating this in the context of NLRP3 and diabetes would increase the cohesiveness.
Response: We have revised the Figure and add some details to demonstrate in the context of NLRP3 and diabetes, as following:
Line 123-132: Figure 1. Activation of NLRP3 and secretion of IL-1β and IL-18. NLRP3 inflammasome consists of the sensor molecule NLRP3, apoptosis-associated speck-like protein containing a CARD (ASC), and pro-caspase-1. There are two steps to activate the NLRP3 inflammasome in macrophages. First, NF-κB expression is induced by inflammatory stimuli (such as TNFα, SFA, or LPS) resulting in the expression of pro-IL-1β and pro-IL-18; Second, caspase-1 mediates the maturation and secretion of IL-1β and IL-18 following activation of the inflammasome by PAMPs and DAMPs. After NLRP3 is activated, cells secrete a large number of pro-inflammatory cytokines (e.g. IL-1β and IL-18), which aggravate glucose tolerance, insulin tolerance and the progression of diabetes. SFA: Saturated fatty acids.
4. Lastly, can the authors please detail how their review differs from PMID: 23483669?
Response: Thanks. We appreciate the positive feedback from the reviewer. We have read the article with PMID: 23483669 and revised the manuscript in response to the extensive and insightful reviewer comments.
Round 2
Reviewer 1 Report
Please, see the attachment. Copy-paste is not working.

Reviewer 3 Report
The authors have incorporated some aspects of the comments, however, the review would further enhance the literature via the incorporation of the following points:
In response to point #1, although the pathological mechanisms of T2DM and intestinal homeostasis have been detailed, their link with the NLRP3 inflammasome, IL-1beta and IL-18 is lacking.
In response to point #2, please incorporate IL-18 in the context of NLRP3 and diabetes, as this has not been addressed.
In response to point #4, please outline the sections of the manuscript where PMID: 23483669 has been incorporated.